# Cross-Species Proteomics Identifies CAPG and SBP1 as Crucial Invasiveness Biomarkers in Rat and Human Malignant Mesothelioma

**DOI:** 10.3390/cancers12092430

**Published:** 2020-08-27

**Authors:** Joëlle S. Nader, Alice Boissard, Cécile Henry, Isabelle Valo, Véronique Verrièle, Marc Grégoire, Olivier Coqueret, Catherine Guette, Daniel L. Pouliquen

**Affiliations:** 1Université de Nantes, Inserm, CRCINA, F-44000 Nantes, France; joelle03nader@gmail.com (J.S.N.); marc.gregoire@inserm.fr (M.G.); 2Université d’Angers, ICO Cancer Center, Inserm, CRCINA, F-44000 Nantes, France; alice.boissard@ico.unicancer.fr (A.B.); cecile.henry@ico.unicancer.fr (C.H.); isabelle.valo@ico.unicancer.fr (I.V.); Veronique.Verriele@ico.unicancer.fr (V.V.); catherine.guette@ico.unicancer.fr (C.G.); 3Université d’Angers, Inserm, CRCINA, F-44000 Nantes, France; olivier.coqueret@univ-angers.fr

**Keywords:** malignant mesothelioma, biomarkers, proteomics, macrophage-capping protein, fatty acid-binding protein, laminin subunit beta-2, selenium-binding protein 1, carcinogenesis

## Abstract

Malignant mesothelioma (MM) still represents a devastating disease that is often detected too late, while the current effect of therapies on patient outcomes remains unsatisfactory. Invasiveness biomarkers may contribute to improving early diagnosis, prognosis, and treatment for patients, a task that could benefit from the development of high-throughput proteomics. To limit potential sources of bias when identifying such biomarkers, we conducted cross-species proteomic analyzes on three different MM sources. Data were collected firstly from two human MM cell lines, secondly from rat MM tumors of increasing invasiveness grown in immunocompetent rats and human MM tumors grown in immunodeficient mice, and thirdly from paraffin-embedded sections of patient MM tumors of the epithelioid and sarcomatoid subtypes. Our investigations identified three major invasiveness biomarkers common to the three tumor sources, CAPG, FABP4, and LAMB2, and an additional set of 25 candidate biomarkers shared by rat and patient tumors. Comparing the data to proteomic analyzes of preneoplastic and neoplastic rat mesothelial cell lines revealed the additional role of SBP1 in the carcinogenic process. These observations could provide new opportunities to identify highly vulnerable MM patients with poor survival outcomes, thereby improving the success of current and future therapeutic strategies.

## 1. Introduction

The management of malignant mesothelioma (MM) remains a challenge today given its complex biology and aggressiveness, and the absence of specific early symptoms [1]. The effect of current and new therapies on overall survival also remains very modest [2], prompting the need to search for biomarkers that could improve early diagnosis, prognosis, and treatment [3]. Sequential Window Acquisition of all Theoretical Mass Spectra (SWATH-MS) has recently emerged as a promising new tool in cancer proteomics, making it possible to identify biomarkers of increasing stages of invasiveness in MM experimental models, for example [4].

Proteomic analyzes of MM have already provided lists of putative cancer biomarkers, although significant differences are observed between primary and commercial MM cell lines [5], for example, emphasizing the need to use best-suited preclinical cellular models [6]. Moreover, long-established human cell lines [7], some genetically engineered mouse models [8], and subcutaneous xenograft models of human tumors [9,10] often fail to predict drug effects in clinical practice. Therefore, to recapitulate the spectrum of tumor heterogeneity seen in patients, and limit the impact of differences in stromal conditions observed between patient and cancer models, cross-species proteomic analyzes are suggested to improve preclinical evaluation [11].

Remembering the importance of potential sources of bias when identifying biomarkers with potential application in oncology [12,13], to determine which invasiveness biomarkers identified in MM experimental models evolved similarly in human MM, we compared lists of proteins of interest from three biological sources. Data were collected firstly from two MM cell lines, secondly from rat MM tumors grown in syngeneic immunocompetent animals and human MM tumors grown in immunodeficient mice, and thirdly from paraffin-embedded sections of patient MM tumors of the epithelioid and sarcomatoid subtypes. The results identified one main biomarker, CAPG, associated with invasiveness and common to all three categories of tumors and human cell lines. Moreover, two other biomarkers were common to the three tumor sources, while 25 other candidates of interest were shared by rat and patient MM tumors. Finally, comparing these data with proteomic analyzes of a large collection of preneoplastic and neoplastic rat mesothelial cell lines revealed the additional role of SBP1 in the carcinogenic process.

## 2. Results

### 2.1. Characterization of Cell Lines and MM Tumors

The four rat MM tumor models shared a sarcomatoid morphology of tumor cells but differed in their infiltrative potential. The M5-T2 tumor is noninvasive, with tumor cell development restricted to the omentum without liver capsular breakthrough (Figure 1A, top left). The F4-T2 tumor is moderately invasive with a regular tumor front (Figure 1A, top right). The F5-T1 and M5-T1 tumors are both characterized by deep infiltration of the liver with irregular tumor fronts, however their respective tumor cells differ in their levels of atypia (Figure 1A, bottom). The highly invasive nature of the M5-T1 tumor is also revealed by necrosis of the liver parenchyma and the presence of apoptotic hepatocytes at the tumor front (Figure 1A, bottom right), associated with the specificities of its proteome [4]. The mean time required for macroscopic tumor development following the injection of 3–5 × 10^6^ cells i.p. into syngeneic rats also differs among the four models: five weeks for M5-T2, four weeks for F4-T2, and three and a half weeks for F5-T1 and M5-T1.

The tumor rate development of the two models of human MM xenografts grown in NOD SCID mice (mice homozygous for the severe combined immune deficiency spontaneous mutation Prkdc^scid^, characterized by an absence of functional T cells and B cells) also differed markedly, with six weeks for MM34 versus three and a half weeks for MM163. These differences were also confirmed at microscopic level, as MM163 was characterized by tumor cells with heterogeneous nuclei in size and shape, prominent nucleoli, the presence of mitotic figures, and frequent atypia (Figure 1B, right) compared with MM 34 (Figure 1B, left).

The two sarcomatoid MM tumors from patients (SMM-1 and S-MM-2) were characterized by abundant intercellular collagen deposition, the presence of spindle-shaped tumor cells with oval nuclei, considerable heterogeneity in cell dimensions, and frequent atypia (Figure 1C, right column). The two epithelioid MM tumors (EMM-1 and EMM-2) contained tumor cells with abundant eosinophilic cytoplasm, round nuclei, and mild nuclear atypia (Figure 1C, left column).

One of the most frequent genomic alterations found in MM concerned *CDKN2A*, observed in the different histologic types [14]. Analysis of mRNA levels of this gene by qRT PCR in cell lines from the two species has previously revealed a comparable decreased relative expression in human pleural MM cell lines relative to normal mesothelial cells, and in rat MM cell lines relative to preneoplastic mesothelial cell lines [15]. Additionally, *Cdkn2a* relative expression was even more decreased in the three invasive MM cell lines (F4-T2: 2.54; F5-T1: 2.10; and M5-T1: 0.79) relative to the non-invasive M5-T2 cell line (4.97) [15]. The bi-allelic deletion of the *CDKN2A* gene, further confirmed in a list of MM human cell lines including the least invasive MM34 (Meso 34), was found to be strongly associated with overexpression of *IL34* and weakly with mutations of the *NF2* gene (with no association with other genetic alterations in *BAP1*, *LATS2* or *TP53* genes) [16]. MM163 (Meso 163) differed from MM34 by a homozygous deletion of the *IFNB1* gene (located in the same 9p21.3 chromosome region as *CDKN2A*) that encodes IFN-β [17]. A transcriptomic analysis of the group of human MM cell lines sharing the same features as MM163, comparing cells exposed to measles virus with untreated cells, revealed these cells were characterized by a weak IFN-I response, some canonical pathways involved in antigen presentation and cytotoxic T lymphocyte-mediated apoptosis of target cells being particularly hit [17]. 

### 2.2. Main Biomarkers Sharing the Same Evolution in the Three Sources of MM Tumors

SWATH-MS data on increased MM tumor invasiveness were collected from (1) comparison of the three invasive rat MM tumors (F4-T2, F5-T1, M5-T1) versus the noninvasive one, M5-T2; (2) comparison of Meso 163 xenografts versus Meso 34 human MM tumors grown in immunodeficient mice; and (3) comparison of human MM tumors from patients, sarcomatoid versus epithelioid subtypes. The main findings are summarized in Table 1. The number of proteins with a fold change > 1.5 and statistical *p*-value < 0.05 estimated by MarkerView was 433, 133, and 191 in each experiment, respectively. Volcano plots for comparisons (1) (2) and (3) are provided in Figure 2A–C, respectively. Comparing these lists, represented by the green, brown, and orange circles, respectively (illustrated in Figure 2D), led us to identify a first pattern of common changes observed in the three experiments and shared by the macrophage-capping protein (encoded by *CAPG*), the fatty acid-binding protein, adipocyte (encoded by *FABP4*), and the laminin subunit beta-2 (encoded by *LAMB2*). These proteins are involved in actin filament finding, lipid transport (fatty acid binding) and extracellular matrix constitution (cell adhesion), respectively. Additional consideration of the comparison of Meso 163 versus Meso 34 human MM cell lines revealed that CAPG was the only biomarker exhibiting similar changes (a strong tendency was also observed for LAMB2), while there were no significant changes for FABP4 (Table 1 and Figure 3). No additional change was observed in the comparison of invasive versus noninvasive rat MM cell lines for the three proteins.

A second pattern of common changes was represented by proteins sharing the same differences between rat and human MM but showing no significant changes in human MM xenografts. Three proteins were involved, poly [ADP-ribose] polymerase 1 (encoded by *PARP1*), vesicle-fusing ATPase (encoded by *NSF*), and inosine-5′-monophosphate dehydrogenase (encoded by *IMPDH2*). These proteins are involved in DNA repair, vesicle-mediated transport (Golgi) and de novo synthesis of guanine nucleotides, respectively. This situation confirms that transplantable tumors established subcutaneously in immunodeficient mice are less relevant in terms of stromal/vascular interactions than orthotopic models of tumors in syngeneic animals [6]. However, these limitations were counterbalanced by the observation of tendencies toward an increase in MM163 vs. MM34 xenografts, while significant differences were also found between corresponding human cell lines (Table 1 and Figure 3).

### 2.3. Additional Biomarkers of Interest Common to Rat and Human MM

Several additional conclusions were drawn from the common changes in abundance limited to rat and patient MM tumors. Firstly, in the 3 versus 1 comparative analysis (Figure 2D), among the 18 proteins exhibiting a common increase, annexin A5 (encoded by *ANXA5*), involved in the blood coagulation cascade (anticoagulant), was the only one showing the same pattern of changes in human MM cell lines (Figure 4). Moreover, three more proteins revealed the same tendency, cytochrome c oxidase subunit 2 (encoded by *MT-CO2*), vesicle-trafficking protein SC22b (encoded by *SEC22B*), and fibronectin (encoded by *FN1*) (Table 1 and Figure 4). These proteins are involved in electron transport (respiratory chain), vesicle-mediated transport (membrane), and extracellular matrix structural composition (cell adhesion and motility), respectively. Finally, among the seven other proteins exhibiting a decreased abundance, two presented the same pattern, selenium-binding protein 1 (encoded by *SELENBP1*), an oxidoreductase also involved in protein transport, and synaptic vesicle membrane protein VAT-1 homolog (encoded by *VAT1*), which negatively regulates mitochondrial fusion (Table 1 and Figure 4). 

The rest of the proteins listed in the 3 versus 1 comparison involved candidate biomarkers for which the difference in abundance between cells was insignificant (*p* > 0.090). By order of magnitude, proteins showing increased abundance with invasiveness included Ras-related protein Rab-31 (encoded by *RAB31*), Ragulator complex protein LAMTOR1 (encoded by *LAMTOR1*), isoform LCRMP-4 of dihydropyrimidinase-related protein 3 (encoded by *DPYSL3*), leucine-rich repeat-containing protein 59 (encoded by *LRRC59*), isoform 2 of tropomyosin alpha-3 chain (encoded by *TPM3*), endoplasmic reticulum resident protein 29 (encoded by *ERP29*), PRA1 family protein 3 (encoded by *ARL6IP5*), ferritin light chain (encoded by *FTL*), isocitrate dehydrogenase [NAD] subunit alpha, mitochondrial (encoded by *IDH3A*), V-type proton ATPase subunit B, brain isoform (encoded by *ATP6V1B2*), and 40S ribosomal protein S18 (encoded by *RPS18*) (Table 1 and Figure 5). Finally, proteins exhibiting a common decrease in both rat and human MM from patients were EH domain-containing protein 2 (encoded by *EHD2*), septin-7 (encoded by *SEPTIN7*), serum albumin (encoded by *ALB*), and two subunits of hemoglobin (encoded by *HBA1* and *HBB*) (Table 1 and Figure 5).

### 2.4. Candidate Biomarkers Common to Xenografts and Rat or Patient MM

Compared with the previous situation (3 versus 1), the numbers of common proteins found in conditions 2 versus 1, and 3 versus 2, were significantly reduced (Figure 2D). Among these lists, the parallel increase in prohibitin (encoded by *PHB*), and decrease in peroxiredoxin-6 (encoded by *PRDX6*) and ezrin (encoded by *EZR*) have previously been reported to be linked to the acquisition of invasive properties in rat MM models [4]. Moreover, these lists contain several candidate invasiveness biomarkers common to MM and other cancer types and reported in the literature, including gelsolin (encoded by *GSN*), profiling-1 (encoded by *PFN1*), glutathione-S-transferase P (encoded by *GSTP1*), keratin, type I cytoskeletal 10 (encoded by *KRT10*), and serpin H1 (encoded by *SERPINH1*) [13].

### 2.5. Abundance Changes during Rat MM Carcinogenesis

We next investigated whether some of the 28 candidate biomarkers (the 18 increased and 7 decreased proteins listed in the 3 versus 1 comparison, plus CAPG, FABP4, and LAMB2) common to the rat and human MM (Figure 2D) exhibited additional abundance changes during the carcinogenesis process. For that purpose, we first examined the SWATH-MS proteomic data of the whole biocollection of rat mesothelial cell lines, looking in particular at the list of 674 proteins differentiating preneoplastic cell lines with sarcomatoid versus epithelioid morphology [18]. In a second step, we compared this list to another list of 192 proteins discriminating the two subgroups of preneoplastic cell lines with sarcomatoid morphology PNsarc2 vs. PNsarc1, which differ in their relative expression of *Hif1a* [18]. Finally, comparing the 94 proteins exhibiting significant abundance changes in the two previous situations with the 28 candidate biomarkers described above (see Figure 2D and Section 2.2 and Section 2.3), led to six proteins common to the four proteomic analyzes (Figure 6A). The absence of FABP4 in this list (the protein was not detected in cells) suggests a location in the stroma. 

Interestingly, among these six proteins, selenium-binding protein 1 (SBP1, encoded by *Selenbp1*) was the only one exhibiting a continuous decrease from the different subgroups of preneoplastic cell lines with epithelioid morphology to PNsarc1 and PNsarc2, including a final additional decrease in neoplastic cell lines (Table 1 and Figure 6D). Conversely, for CAPG and RAB31, protein abundances in neoplastic cells differed significantly from only one of the two groups of preneoplastic cell lines (Table 1 and Figure 6B,C). For comparison, proteomic data for LAMB2 revealed the absence of significant changes within the different groups and subgroups of preneoplastic cell lines, while there was a dramatic decrease in all neoplastic cell lines (Table 1 and Figure 6E). For fibronectin, the evolution of abundance showed a progressive rise within the first four subgroups of preneoplastic cell lines but as above discrimination with neoplastic cells was incomplete (Table 1 and Appendix A). Finally, for TPM3 and VAT1, no clear evolution was observed within the different groups and subgroups of preneoplastic cells in comparison with neoplastic cells (Table 1 and Appendix A).

## 3. Discussion

This study investigated the proteomic changes associated with MM invasiveness that were common to experimental and human cell lines or tumor models generated in the F344 rat, human tumor xenografts, and tumor specimens from patients. Our investigations identified three major invasiveness biomarkers not documented so far in integrative molecular studies characterizing MM [14], common to the three tumor sources, CAPG, FABP4, and LAMB2, and an additional set of candidate biomarkers shared by rat and patient tumors. Among these, SBP1 appeared to play an additional crucial role in the carcinogenic process of mesothelial cells.

CAPG, together with ANXA5 and FABP4, was previously found within a group of biomarkers differentiating invasive from noninvasive MM rat tumor models, their abundance being very significantly increased and decreased, respectively [4]. This actin filament end-capping protein was initially reported to be increased in the transformation of human breast cancer cells into a highly metastatic variant [19]. Herein, we confirm that CAPG is also increased in human MM cell lines, human MM tumor models, and patient MM. Interestingly, our observations are consistent with several previous reports showing this protein’s overexpression in different cancer types. Its role in promoting the invasiveness of cholangiocarcinoma and hepatocellular carcinoma has been established by Morofuji et al. [20], and Kimura et al. [21], respectively. Its involvement in migration and invasiveness has been documented for ovarian carcinoma by Glaser et al. [22], and for breast cancer by Davalieva et al. [23] and Huang et al. [24]. Its upregulation in clinical high-grade glioblastoma has also been reported by Xing and Zeng [25], while the correlation of its expression level with shorter survival time was demonstrated by Fu et al. [26]. Moreover, the link between its abundance and occurrence of lymph node metastasis has also been documented for three different types of cancer [20,27,28], as well as its association with the prediction of response to treatment [20,29].

FABP4 (also called A-FABP or aP2) is 1 of 10 members of a family of proteins involved in intracellular fatty acid transport and lipid trafficking regulation in cells, which show different tissue-specific expression patterns [30]. Its previously mentioned adipokine function regulating macrophage and adipocyte interactions during inflammation [31] may be consistent with the absence of significant differences observed in our study between mesothelial and MM cell lines. We previously reported that the extent of the decrease was related to increasing invasiveness in rat MM [4]. Interestingly, our observations also agree with the findings of Mathis et al. showing that FABP4 loss was associated with high stage/grade and the presence of metastatic lymph nodes in invasive bladder cancer [32]. Zhong et al. have also demonstrated that similar observations are made in hepatocellular carcinoma, with the protein’s overexpression leading to tumor growth inhibition in vivo [33]. A second common protein exhibiting a decreased abundance in all tumor sources was laminin subunit beta-2 (LAMB2). This protein belongs to a family of 16 laminin isoforms, which combine with subunits of collagen IV to build the basement membranes surrounding blood vessels, lymphatics, nerves, and muscle cells. Hewitt et al. initially reported that within carcinomas, vascular basement membrane staining for the subunit beta-2 is clearly weaker relative to normal tissues, probably due to their incomplete maturation [34]. This observation was further confirmed by immunohistochemistry by Mustafa et al. when studying angiogenesis in glioblastoma [35]. The fascinating aspects of their structural diversity have been emphasized by Hohenester and Yurchenco [36], raising crucial questions on the challenge that studying their complex interactions in vivo presents.

The first of an additional subset of common biomarkers of interest, which differed from the previous three by the absence of significant changes in xenografts (only a tendency), was represented by PARP1. The recent development of PARP1 inhibitors for the treatment of cancers presenting compromised HR repair has led to interesting findings on biomarkers associated with their clinical use against MM [37]. Moreover, Gaetani et al. revealed the relationship between PARP1 and miR-126 regulation in the context of asbestos-induced malignancy [38]. Regarding NSF, changes have not yet been documented in the context of cancer invasiveness; however. our data suggest that the increase commonly observed is related to the reassembly pathway of Golgi cisternae at the end of mitosis [39]. Finally, our results are consistent with the recent finding by Kofuji et al. that overexpression of the rate-limiting enzyme for de novo guanine nucleotide biosynthesis, IMDH2, relative to primary glia, promotes glioblastoma tumorigenesis [40]. Among the other biomarkers for which no changes were observed in xenografts, the most significant differences in abundance were found for annexin A5. The potential of the smallest member of the annexin family as a predictive biomarker for tumor development, metastasis, and invasion has already been reviewed [41], with it also being involved in cell membrane repair [42]. Our results are consistent with reports of its overexpression in several other cancer types, including renal cell carcinoma [43], colon cancer [44], and hepatocarcinoma [45,46]. Other highly significant changes mainly involve two proteins, COX2 for increase and VAT1 for decrease. Cytochrome c oxidase dysfunction has already been demonstrated to be related to the Warburg effect in invasive cancers [47]. The involvement of VAT1, a largely uncharacterized enzyme, has also been reported in the regulation of cancer cell motility and its interaction with Talin-1, a key cytoskeletal protein [48].

Two other proteins caught our attention among the second additional subset of common biomarkers of interest, EHD2 and RAB31, characterized by highly significant changes in abundance in both rat and patient MM, but not in human cell lines or in xenografts. The level of the first protein, which belongs to the EHD family associated with plasma membrane, has been reported to be reduced in human esophageal squamous carcinoma in comparison with adjacent normal tissues, linked to increased motility of the tumor cells [49]. Subsequently, a decreased expression was also observed, correlated with histological grade, in an immunohistochemical study of 96 human breast carcinoma samples, leading Shi et al. to suggest that this protein inhibits metastasis by regulating EMT [50]. The second protein, which belongs to the small GTPase family Rab and to the Rab5 subfamily, presents an estrogen receptor-responsive element in its promoter region which can be dysregulated in breast cancer cells, the consequences of this key finding in cancer research having been reviewed by Chua and Tang [51].

Both CAPG and RAB31 shared a similar pattern of changes during the course of rat mesothelial cell carcinogenesis. However, these changes were only observed in the first two subgroups of preneoplastic cell lines with sarcomatoid morphology, suggesting a link to increased *Hifa* expression [18]. The pattern of changes observed for SBP1 markedly contrasted with these situations as decreases in abundance were observed at three main stages of the carcinogenic process. Firstly, the decrease observed between PNep and PNint was concomitant with the first dramatic decrease in the expression of *Cdh1* and *Il10*, and parallel increase in the expression of *Acta 2*, *Tgfb1* [15]. Secondly, the new decrease observed between PNsarc1 and PNsarc2, and continuous decrease from PNep to PNsarc2, confirm the existence of links to both the level of expression of *Hifa* [18] and EMT process [15]. Thirdly, the decrease observed between preneoplastic cell lines with both epithelioid and sarcomatoid morphologies and neoplastic cell lines leads to the conclusion that SBP1 presents additional interest as a biomarker of neoplastic transformation. Finally, the decrease in SBP1 also observed in association with increased invasiveness in human cell lines, rat and patient MM tumors tends to confirm the protein’s crucial role. The downregulation of another selenium-containing protein was earlier reported by Apostolou et al., suggesting that selenium could be useful as a chemopreventive agent in individuals at high risk of MM due to asbestos exposure [52]. Interestingly, Rundlöf et al. found differential expression within isoforms of the selenoenzyme thioredoxin reductase 1 (TrxR1) in MM cell lines, with the sarcomatoid phenotype showing the lower total TrxR1 mRNA level [53]. The mechanisms by which dietary selenium may affect MM tumor progression have only been partly explored, mostly in cell lines, pointing to the crucial role of redox metabolism [54]. Although it is well established that low levels of SBP1 are frequently associated with poor clinical outcome [55], the complexity of selenium metabolism has highlighted the fact that among selenocysteine-containing proteins that are members of the glutathione peroxidase family, SBP1 is the only one for which no catalytic function has been assigned [56]. Therefore, many aspects of this research field require further investigation. To give just a few more very recent examples of the protein’s importance, Lee et al. have suggested that hepatitis B virus-X-expressing cells, which show markedly decreased *SELENBP1* expression, might be one factor in the development of hepatocellular carcinoma caused by HBV infection [57]. Wang et al. have also reported this protein’s novel function in transcriptionally modulating p21 expression through a p53-independent mechanism, with a resulting impact on the G_0_/G_1_ phase cell cycle arrest in bladder cancer [58].

## 4. Materials and Methods

### 4.1. Study Approval

The human studies were conducted according to the ethical guidelines of the Declaration of Helsinki. The paraffin-embedded human MM tumor pieces were prepared from samples of the Tumor Bank of the Reims University Hospital Biological Resource, Collection No. DC-2008-374, declared to the Ministry of Health according to French law, for the use of tissue samples for research. The two human cell lines MM34 (Meso 34) and MM163 (Meso 163) were established from pleural effusions of patients with suspected pleural MM [59], according to the ethics committee approval (Comité de Protection des Personnes Ouest IV-Nantes, dossier n° DC-2011-1399). The animal studies were carried out in agreement with European Union guidelines for the care and use of laboratory animals in research protocols (Agreement #01257.03). All experiments were approved by the ethics committee for animal experiments of the Pays de la Loire Region, France (CEEA.2011.38 and CEEA.2013.7.). 

### 4.2. Rat and Human Cell Lines, and Tumor Samples

The 27 cell lines of the rat biocollection were grown in RPMI 1640 medium, supplemented with 10% heat-inactivated fetal calf serum, 2 mM L-glutamine, 100 U/mL penicillin, and 100 µg/mL streptomycin (all reagents from Gibco Life Technologies Limited, Paisley, UK) at 37 °C in a 5% CO_2_ atmosphere. Cells were collected from preconfluent 75 cm^2^ flasks and cell pellets of 2 × 10^6^ cells were used for SWATH-MS proteomic analysis after washing in PBS buffer. The four rat neoplastic cell lines (M5-T2, F4-T2, F5-T1, and M5-T1) were injected into syngeneic rats, and tumors collected and fixed as previously described [4]. The two human cell lines Meso 34 and Meso 163 were established from pleural effusions of patients with suspected pleural MM, aseptically collected by thoracocentesis as previously described [56], and cultivated as rat cell lines. Meso 34 and Meso 163 xenografted tumors were collected and fixed after injection of the corresponding cell lines into the peritoneal cavity of two groups of five immunodeficient NOD SCID mice. For patient tumors, four pieces of paraffin-embedded pleural MM tumor pieces collected from four different patients were obtained from the Tumor Bank of the Reims University Hospital Biological Resource. They represented two tumors of the sarcomatoid subtype (S-MM1 and S-MM2) versus two tumors of the epithelioid subtype (E-MM1 and E-MM2).

### 4.3. SWATH-MS Analysis

The spectral libraries, DDA experiments, peptide identification, and peak extraction of the SWATH data were performed as previously described [4], using either Spectronaut software (v 8.0, Biognosys, Schlieren, Switzerland) or the SWATH micro app embedded in PeakView (v 2.0, AB Sciex Pte. Ltd., Framingham, MA, USA). Sections of the tumors, stained with hematoxylin-phloxine-saffron (HPS), were first examined to select areas of interest, then removed with a scalpel. Five 20 µm thick sections of the samples were used, and the areas of interest collected in a microtube. Samples were deparaffinized, and then cell pellets and dried deparaffinized tumor samples treated as previously described [4]. After centrifugation, salts were removed using OASIS^®^ HLB extraction cartridges (Waters SAS., St Quentin-en-Yvelines, 78, France), and the samples dried under SpeedVac. Peptide concentrations of the samples were determined using the Micro BCA^TM^ protein assay kit (Thermo Fisher Scientific, St Herblain, 44, France). 

Five micrograms of each sample were analyzed with a SWATH-MS acquisition method. The method of acquisition, peak extraction of the SWATH data, calibration of the retention time of extracted peptide peaks and quantification followed the procedure already described in [4]. For statistical analysis of the SWATH data set, the peak extraction output data matrix from PeakView was imported into MarkerView (v 2, Sciex, Framingham, MA, USA) for data normalization and relative protein quantification. Proteins with a fold change >1.5 and statistical *p*-value < 0.05 estimated by MarkerView were declared differentially expressed under different conditions.

## 5. Conclusions

This study pointed to some proteins of interest that exhibited the same patterns of quantitative changes in different situations, and for which the relationship with tumor invasiveness has already been reported in the literature for other cancer types. Although this study was limited by the small number of samples, an interesting point was the similarity of observations made on malignant mesothelioma cells and tumors from different sources and from two different species. Extending these studies to a larger number of samples would be the logical next step, which may later contribute to improving current therapies for patients with the worst survival outcomes. Another interesting prospect is related to the questions raised by the additional involvement of the selenium-binding protein 1 in the carcinogenic process, a point that would present a good basis for further basic research in cancerology, and probably also for improving early MM diagnosis.

## Figures and Tables

**Figure 1 cancers-12-02430-f001:**
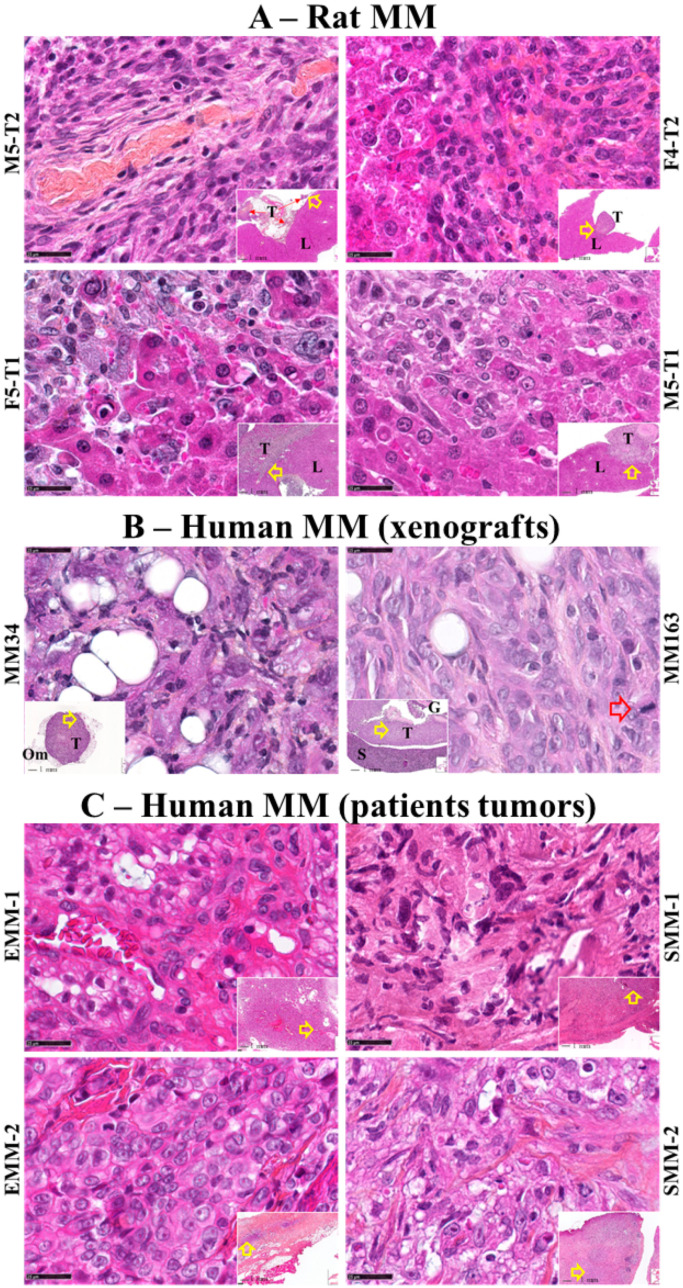
Histological features of the three sources of malignant mesothelioma MM tumors. High magnification views, hematoxylin-phloxine-saffron (HPS) staining (×800, scale bars represent 25 µm), and general views in inserts (×25, scale bars represent 1 mm) with open red arrows indicating the location of magnifications. (**A**) Rat MM tumors of the four experimental models (the names of the corresponding cell lines are indicated on the external side of the photographs). These representative tumor (T) histological sections included liver tissue (L) and tumor cells exhibiting increasing levels of invasiveness. (**B**) Xenografts of human MM tumors grown in NOD SCID mice, with the corresponding cell line names indicated on the external side of the photographs. (Om) = omentum, (G) = gut, (S) = spleen. The large open arrow shows a mitotic figure. (**C**), Human MM tumors from patients. EMM-1 and EMM-2 (left column) = epithelioid histotype, SMM-1 and SMM-2 (right column) = sarcomatoid histotype.

**Figure 2 cancers-12-02430-f002:**
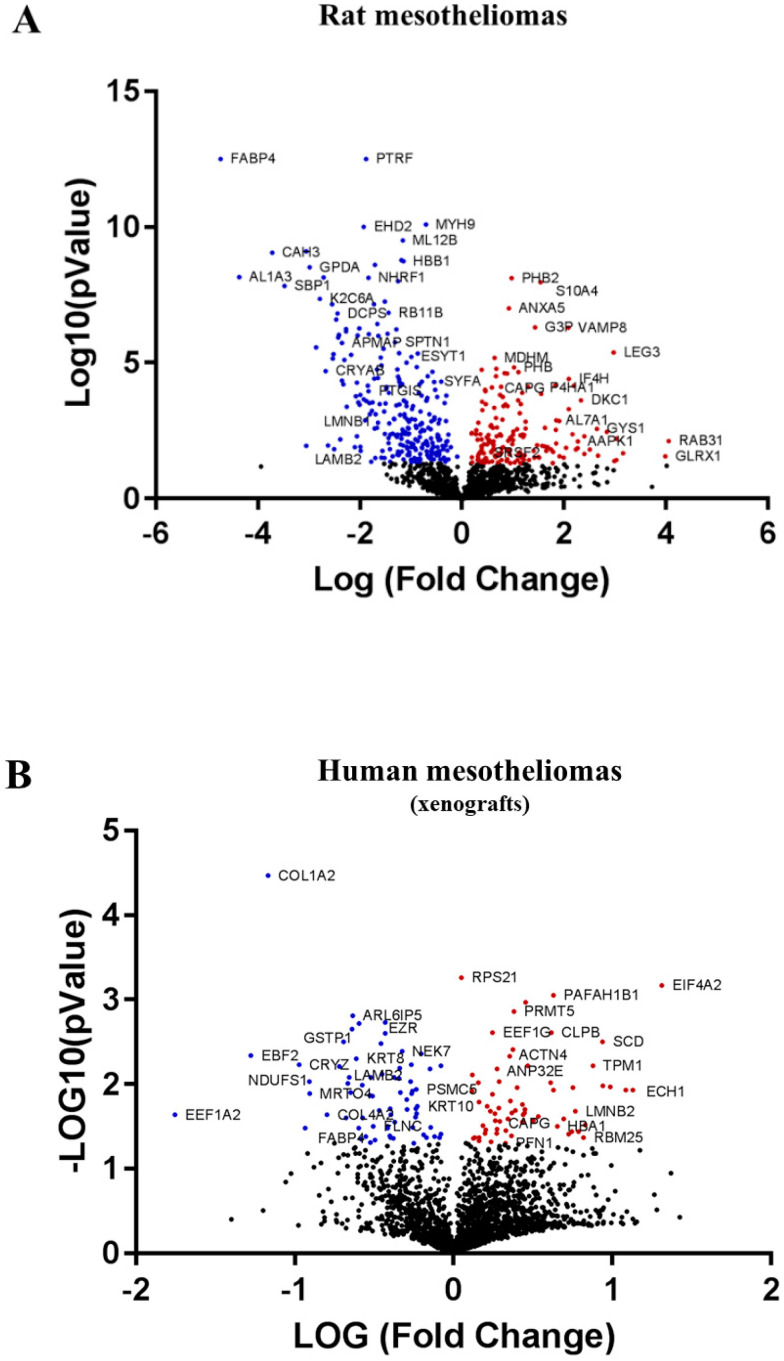
Volcano plots and schematic representation of the comparative proteomic analyzes. (**A**), Volcano plot of the comparison of the three invasive rat MM tumors (F4-T2, F5-T1, and M5-T1) versus the noninvasive one, (M5-T2). (**B**), Volcano plot of the comparison of Meso 163 versus Meso 34 xenografts of human MM tumors grown in immunodeficient mice. (**C**), Volcano plot of the comparison of human MM tumors from patients, sarcomatoid versus epithelioid subtypes. The locations of CAPG (in red), FABP4 and LAMB2 (in blue) are indicated in the three volcano plots. (**D**), Schematic representation of the comparative proteomic analyzes. The three different sources of MM tumors are illustrated by the green (Rat MM), brown (xenografts of human MM grown in NOD SCID mice) and orange (human MM from patient tumor samples) circles. The green circle represents the 433 proteins showing significant changes in abundance (*p* < 0.05) between the three invasive rat MM tumors versus the noninvasive one. The brown circle illustrates the 133 proteins showing significant changes in abundance (*p* < 0.05) between Meso 163 (MM163) and Meso 34 (MM34) xenografts. The orange circle represents the 191 proteins affected by significant changes in abundance (*p* < 0.05) between the two sarcomatoid versus the two epithelioid MM tumors from patients. Genes coding for proteins exhibiting common significant changes are given for *homo sapiens* in italics (increase in red, decrease in blue).

**Figure 3 cancers-12-02430-f003:**
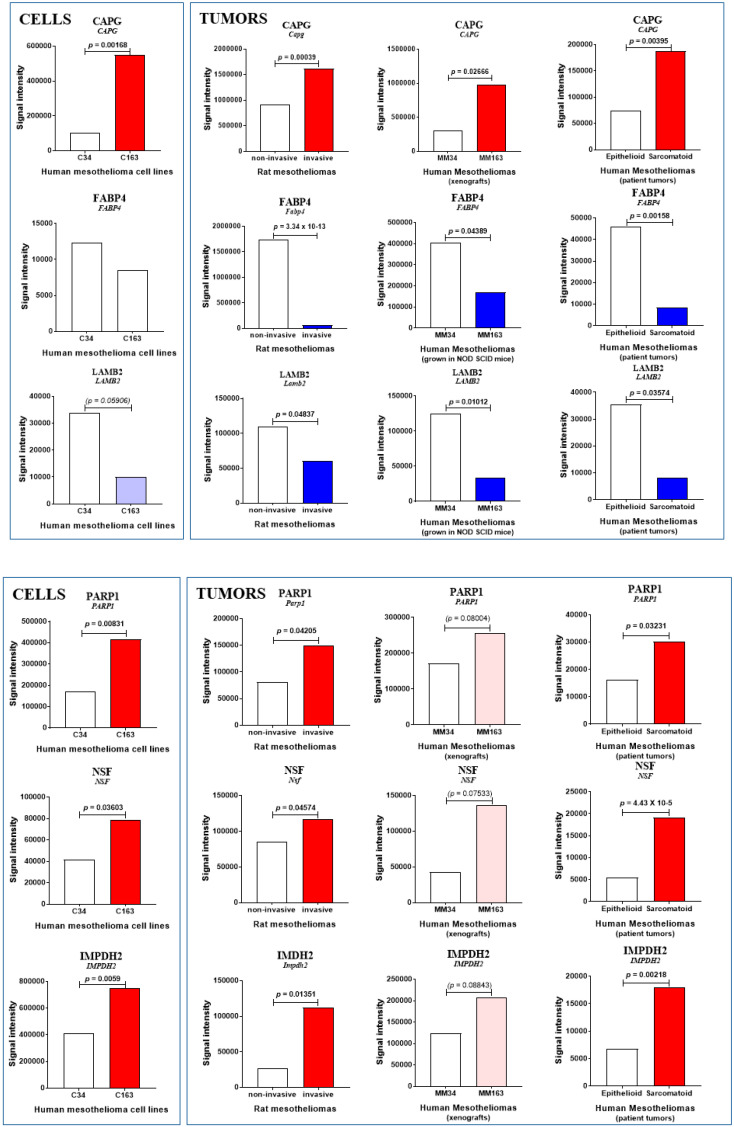
Common biomarkers of MM invasiveness. Proteins showing comparable abundance changes in MM from the three sources and between human mesothelioma cell lines. Increase and decrease are indicated by red and blue bars, respectively (with *p* values). Blank bars reflect the absence of significant changes (*p* > 0.09), while light red or blue bars indicate tendencies (0.05 < *p* < 0.09).

**Figure 4 cancers-12-02430-f004:**
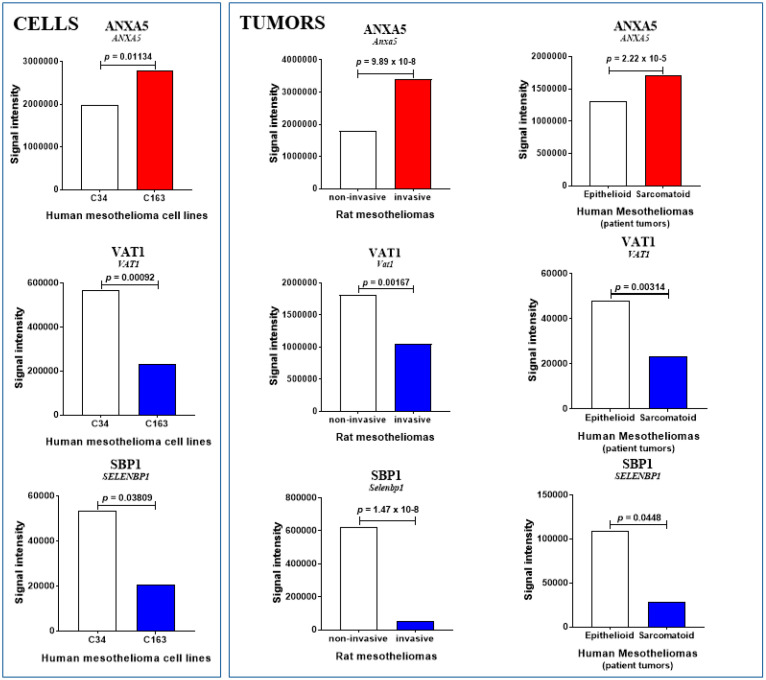
Main invasiveness biomarkers in human vs. rat MM, and cell lines. Proteins showing comparable abundance changes in MM from rat models and patients, and human mesothelioma cell lines. Increase and decrease are indicated by red and blue bars, respectively (with *p* values). Blank bars reflect the absence of significant changes (*p* > 0.09), while light red bars indicate tendencies (0.05 < *p* < 0.09).

**Figure 5 cancers-12-02430-f005:**
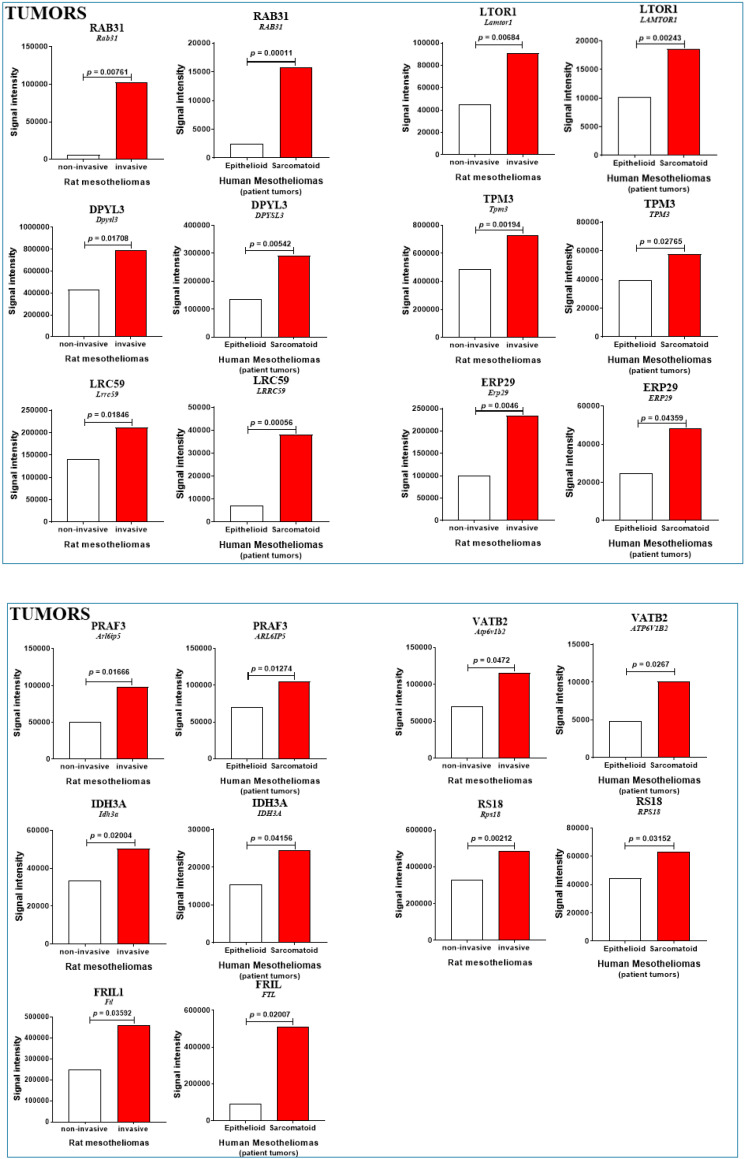
Additional invasiveness biomarkers in MM from patient and rat models. Proteins showing comparable abundance changes only in patient MM vs. rat models. Increase and decrease are indicated by red and blue bars, respectively (with *p* values). Blank bars reflect the absence of significant changes (*p* > 0.09). For clarity, data on the beta subunit of hemoglobin (encoded by *HBB*) have been excluded as they were similar to those observed for the alpha subunit (encoded by *HBA*).

**Figure 6 cancers-12-02430-f006:**
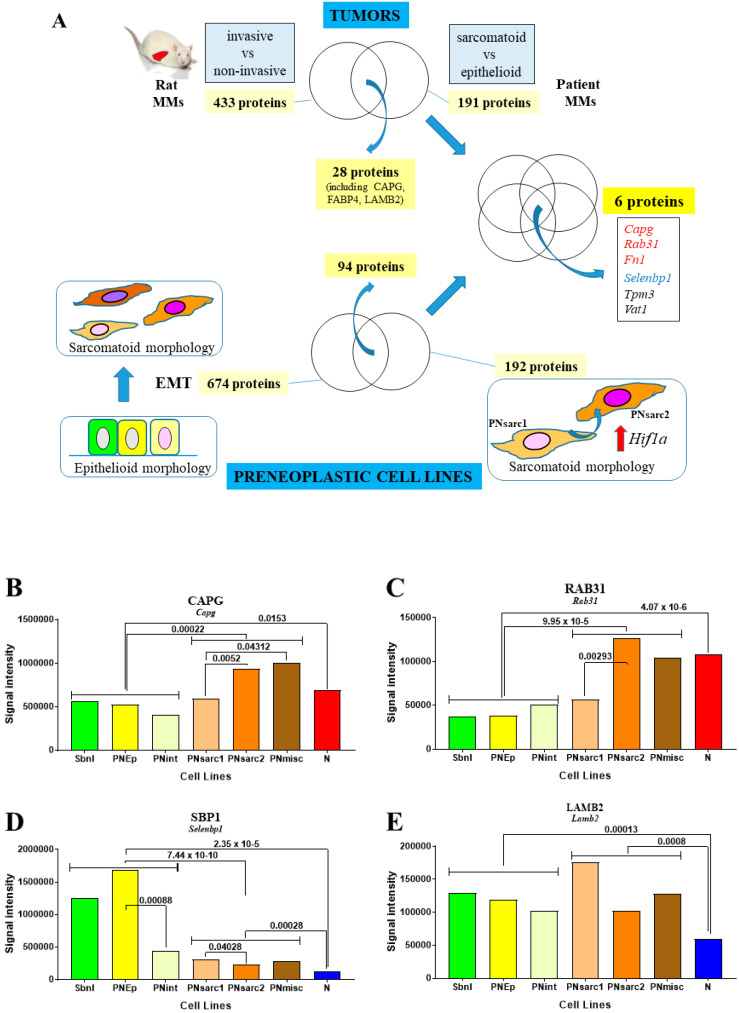
Biomarkers of human vs. rat MM and rat mesothelial cell carcinogenesis. (**A**), Diagram of the methodology used to identify biomarkers showing additional changes during the course of rat mesothelial cell carcinogenesis. For both CAPG (**B**) and RAB31 (**C**), a common rise in abundance was specifically observed in PNsarc2 vs. PNsarc1 and between the whole groups of preneoplastic cell lines with sarcomatoid vs. epithelioid morphology. (**D**), Evolution of abundance changes for SBP1. (**E**), Evolution of abundance changes for LAMB2.

**Table 1 cancers-12-02430-t001:** Summary of proteomics biomarkers of MM invasiveness and carcinogenesis. Abundance changes: + *p* < 0.05; - ns (*p* > 0.09); (+) tendency (0.05 < *p* < 0.09).

Protein	Rat MM	Patient MM	Human Xenografts	Human MM Cell Lines	Rat MM Carcinogenesis
CAPG	+	+	+	+	+/−
FABP4	+	+	+	−	−
LAMB2	+	+	+	(+)	−
PARP1	+	+	(+)	+	−
NSF	+	+	(+)	+	−
IMDH2	+	+	(+)	+	−
ANXA5	+	+	−	+	−
VAT1	+	+	−	+	+/−
SBP1	+	+	−	+	+
COX2	+	+	−	(+)	−
SC22B	+	+	−	(+)	−
FINC	+	+	−	(+)	+/−
RAB31	+	+	−	−	+
DPYL3	+	+	−	−	−
LRC59	+	+	−	−	−
LTOR1	+	+	−	−	−
TPM3	+	+	−	−	+/−
ERP29	+	+	−	−	−
PRAF3	+	+	−	−	−
IDH3A	+	+	−	−	−
FRIL1	+	+	−	−	−
VATB2	+	+	−	−	−
RS18	+	+	−	−	−
EHD2	+	+	−	−	−
SEPT7	+	+	−	−	−
ALBU	+	+	−	−	−
HBA	+	+	−	−	−
HBB	+	+	−	−	−

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
