# Peer review of "Cross-Species Proteomics Identifies CAPG and SBP1 as Crucial Invasiveness Biomarkers in Rat and Human Malignant Mesothelioma"

_cancers, 2020, doi:10.3390/cancers12092430_

Round 1

Reviewer 1 Report

In the manuscript titled, “Cross-species proteomics identifies CAPG and SBP1 1 as crucial invasiveness biomarkers in rat and human 2 malignant mesothelioma”, the authors have mass-spectrometry proteomics profile of cell-line, xenograft models, and tissue sample of malignant mesothelioma (MM) to identify biomarker proteins associated with MM invasiveness. Comparing the protein expression between the samples used the authors identified a set of biomarkers including CAPG, FABP4, LAMB2 associated with invasiveness.

Overall, the study the good and the manuscript is written well. However, I have following comments.

  1. As a baseline characterization of the cell-line, xenograft models, and tissue sample used in the study, it would be informative if the authors demonstrate the status of proteins (protein expression and/or IHC) such as BAP1, NF2, TP53, LATS2, SETD2, SETDB1 that are known to play vital role in MM development.
  2. Line 33: Provide full of SWATH-MS.
  3. Figure 2: Volcano plots of the comparisons 1,2, and 3 to show consistency of the foldchange of the proteins highlighted Venn-diagram would be a good addition to Figure 2.
  4. The authors have mostly focused on overlap of proteins between different analysis. What are common enriched signaling pathways common between the analysis performed.
  5. The authors have identified a set of biomarkers including CAPG, FABP4, LAMB2 associated with invasiveness using different cell-line, xenograft models, and tissue sample. How does the mRNA expression patten of these biomarkers differ between the epithelioid and sarcomatoid subtypes of MM in TCGA cohort (PMID: 30322867).

Author Response

We thank very much reviewer # 1 for important and useful comments and suggestions which allowed us to improve the quality of our manuscript.

Please find below our responses to the different points:

1. An important paragraph has been added to the text in section 2.1 of Results and a new title is provided for the same purpose (lines 59, and 81 to 96). We refer to several important data already published regarding CDKN2A and other genes of interests for our rat and human MM cell lines (used for xenografts), as it represents the most frequent tumor suppressor gene homozygously deleted (36/74 mentioned in Hmeljak et al., ref. [14] added in the text, also in link with point 5. of your comments). Unfortunately, we could not have access to equivalent data for MM tumor from patients.

2. The full name of the abbreviation is included in section 1. of the text (lines 34-35).

3. A volcano plot of the comparison 1 (rat MMs, invasive vs non-invasive) is provided in the new Fig. 2 (Top), which represented the basis of other further comparisons with 2 (human xenografts) and 3 (patients tumors). It corresponded to the largest list of proteins analyzed. Additional sentence is also included in the text in section 2.2 (lines 104-105). Given the short delay given for revision of the manuscript, it was impossible to provide the same for comparisons 2 and 3 as colleagues in charge of this task in our laboratory are presently in holidays.

4. Additional information on this point was included in the text for the most important biomarkers described (exhibited in figures 3 and 4), lines 110-111 and 121-122 (section 2.2), and lines 183-184, 187-189, and 191- 193 (section 2.3).

5. We are sorry, we could not find any data on CAPG, FABP4 or LAMB2 in the reference mentioned (added in the text at the beginning of the discussion, lines 276-277). As it represents a huge amount of crucial data and impressive work on the molecular characterization of MM, this reference has been added as [14] in the text (lines 81-82). See also response to point 1. (above).

Reviewer 2 Report

I would like to congratulate authors for this interesting paper. A clear and well-conducted study on the proteomics and biomakerse of Mesothelioma represents an excellent starting point to address further specific research lines, eventually more clinical. 

I found the article very interesting but, if possible, I suggest to summarize all the findings in a more clear table in order to highlight the significant results.  

Author Response

We thank very much reviewer # 2 for comments and suggestions, please find below our responses:

As suggested, we have added a table (Table 1) summarizing the main findings and included in the manuscript (line 137) with corresponding sentences / words (lines 102, 114, 127, 187, 193, 211, 215, 252, 254, 256 and 261).